# Changes in the Incidence of Infantile Spinal Muscular Atrophy in Shikoku, Japan between 2011 and 2020

**DOI:** 10.3390/ijns8040052

**Published:** 2022-09-26

**Authors:** Kentaro Okamoto, Hisahide Nishio, Takahiro Motoki, Toshihiro Jogamoto, Kaori Aibara, Yoichi Kondo, Kentaro Kawamura, Yukihiko Konishi, Chiho Tokorodani, Ritsuo Nishiuchi, Mariko Eguchi

**Affiliations:** 1Department of Pediatrics, Ehime Prefectural Imabari Hospital, 4-5-5 Ishi-cho, Imabari 794-0006, Japan; 2Department of Occupational Therapy, Faculty of Rehabilitation, Kobe Gakuin University, 518 Arise, Ikawadani-cho, Nishi-ku, Kobe 651-2180, Japan; 3Department of Community Medicine and Social Healthcare Science, Kobe University Graduate School of Medicine, 7-5-1 Kusunoki-cho, Chuo-ku, Kobe 650-0017, Japan; 4Department of Pediatrics, Ehime University Graduate School of Medicine, Shitsukawa, Toon 791-0295, Japan; 5Department of Pediatrics, Matsuyama Red Cross Hospital, 1 Bunkyo-cho, Matsuyama 790-8524, Japan; 6Toseikai Healthcare Corporation, Life-Long Care Clinic for Disabled People, 14-3-10 Maeda 4 jo, Teine-ku, Sapporo 006-0814, Japan; 7Department of Pediatrics, Faculty of Medicine, Kagawa University, 1750-1 Ikedo, Miki-cho, Kita 761-0701, Japan; 8Department of Pediatrics, Kochi Health Sciences Center, 2125-1 Ike, Kochi 781-8555, Japan

**Keywords:** infantile spinal muscular atrophy, incidence, epidemiological study, Shikoku, Japan

## Abstract

Spinal muscular atrophy (SMA) is an autosomal recessive neuromuscular disorder. Al-though there was no cure for SMA, newly developed therapeutic drugs (nusinersen, onasemnogene abeparvovec, and risdiplam) have been proven effective for the improvement of motor function and prevention of respiratory insufficiency of infants with SMA. Nusinersen was introduced in Japan in 2017 and onasemnogene abeparvovec in 2020. We hypothesized that the introduction of these drugs might influence the incidence of SMA (more precisely, increase the diagnosis rate of SMA) in Japan. To test this hypothesis, we conducted a second epidemiological study of infantile SMA using questionnaires in Shikoku, Japan between October 2021 and February 2022. The incidence of infantile SMA during the period 2016–2020 was 7.08 (95% confidence interval [CI] 2.45–11.71) per 100,000 live births. According to our previous epidemiological study, the incidence of infantile SMA during 2011–2015 was 2.70 (95% CI 0.05–5.35) per 100,000 live births. The increased incidence of infantile SMA suggests that the widespread news in Japan regarding the introduction of therapeutic agents, nusinersen and onasemnogene abeparvovec, raised clinicians’ awareness about SMA, leading to increased and earlier diagnosis of SMA in Shikoku.

## 1. Introduction

Spinal muscular atrophy (SMA) is an autosomal recessive neuromuscular disorder, resulting in muscle weakness and paralysis [1]. SMA is currently divided into five clinical subtypes according to clinical severity: types 0 to 4 [2]. In type 0, patients have severe muscle weakness and respiratory failure at birth; these patients only survive a few weeks. Type 1 involves onset before age 6 months, with patients rapidly developing limb weakness and respiratory distress; these patients are unable to sit unassisted. Type 2 involves onset at age 6–18 months; patients are able to sit unassisted but cannot stand or walk independently. In SMA types 3 and 4, patients are able to stand or walk independently. However, the ages at onset are different; in type 3, patients develop symptoms in childhood whereas patients with type 4 develop symptoms in adulthood. In this study, “infantile SMA” is used to denote types 0, 1, and 2.

The *survival motor neuron 1* (*SMN1*) is an SMA-determining gene [3]. *SMN1* is absent in 95% of patients with SMA [3]. The remaining patients carry an intragenic mutation in the retaining *SMN1* [3,4]. The number of copies of *survival motor neuron 2* (*SMN2*) is related to severity of the disease. More copies of *SMN2* is related to an improved prognosis [2,4,5]. In general, nearly all patients with SMA type 1 have one or two copies of *SMN2*. Most patients with SMA type 2 have three copies of *SMN2*, and most with SMA type 3 have three or four copies of *SMN2* [2,5,6]. Patients with SMA type 4 usually have four copies or more [2]. However, this inverse correlation is not absolute; some patients with two copies of *SMN2* have milder phenotypes whereas some with three copies of the gene have been described as having SMA type 1 [5].

SMA is a rare disease, and the incidence of SMA varies among countries [7]. Reports of SMA incidence before 1995 were based on clinical symptoms; however, after 1995, reports of SMA incidence were mainly based on genetic testing [6]. Currently, the number of SMA incidence research reports based on newborn screening (NBS) is increasing [8]. Even in the era of NBS, the incidence of SMA continues to change owing to many factors, including awareness about the disease among the population [9]. It should be noted that the current NBS program does not include detection of an intragenic mutation in *SMN1* because the occurrence of intragenic mutation in *SMN1* is very rare.

Although SMA is considered an incurable disease, new drugs for SMA have been available in Japan since 2017 [6], with nusinersen (Spinraza^®^) approved in 2017, onasemnogene abeparvovec (Zolgensma^®^) in 2020, and risdiplam (Evrysdi^®^) in 2021. Clinical trials have demonstrated that these drugs improve the motor function of patients with SMA [10,11,12]. Early treatment with nusinersen or onasemnogene abeparvovec, especially in the pre-symptomatic stage, has been reported to show maximum effects on the improvement of motor function [13,14,15]. For the early diagnosis and treatment of SMA, NBS for SMA has been increasing worldwide [8].

The introduction of new, effective drugs may raise awareness about SMA among clinicians. According to our previous study based on genetic testing [16], the incidence of infantile SMA was 2.70 (95% confidence interval [CI] 0.05–5.35) per 100,000 live births in Shikoku, Japan during the period 2011–2015. This result was much lower than expected, considering reports from other countries, which might be owing to low levels of awareness about SMA among clinicians in Shikoku.

As mentioned, nusinersen was introduced in Japan in 2017 and onasemnogene abeparvovec in 2020. We therefore hypothesized that the introduction of nusinersen influenced the incidence of SMA (more precisely, increased the diagnosis rate of SMA) in Shikoku. To test this hypothesis, we conducted a second epidemiological study of infantile SMA using questionnaires in Shikoku, Japan, between October 2021 and February 2022.

## 2. Materials and Methods

### 2.1. Research Area and Population

This study was conducted in the same area as in our previous report: Shikoku, Japan [16]. Shikoku is one of the largest islands of Japan, separated from the main island by the Seto Island Sea. Shikoku comprises four prefectures: Ehime, Kagawa, Tokushima, and Kochi. Demographic data were obtained from the Portal Site of Official Statistics of Japan [17]. The residents of Shikoku account for approximately 3% of the Japanese population.

This epidemiological study was approved by the Institutional Review Board of Ehime University Hospital (approval number: 1610003, 24 October 2016).

### 2.2. First and Second Questionnaires

A first questionnaire was sent to 84 hospitals with a pediatric department in Shikoku between October 2021 and February 2022. We queried whether any patients with SMA were born in Shikoku between 2016 and 2020 (Appendix A).

A second questionnaire was sent to hospitals that reported having patients with SMA. The questionnaire asked about patient information, including date of birth, sex, clinical subtype, respiratory condition, *SMN1* and *SMN2* copy numbers, and therapeutic agents (Appendix A). Prior to collecting patient information, informed consent was obtained from the parents of patients with SMA.

### 2.3. Statistical Analysis

The SMA incidence is presented as patient number per 100,000 live births. The 95% CIs of the incidence were calculated based on the Poisson distribution, using Microsoft Excel for Mac version 16.63 (Microsoft Corporation, Redmond, WA, USA).

## 3. Results

### 3.1. Demographic Data of Shikoku

The total population in Shikoku and in Japan during 2011–2020 was 3,697,000 and 126,146,000, respectively (Appendix A). The number of live births in 2011–2020 is shown in Appendix A. Both the population and number of live births decreased between 2011 and 2020; the total of live births in Shikoku was 147,950 in 2011–2015 and 127,092 in 2016–2020.

### 3.2. Patients Identified between 2016 and 2020

In this case, 81 hospitals (96.4%) responded to the first questionnaire; nine patients with SMA were newly identified in this study: six male and three female patients. All patients were born in Shikoku in the period 2016–2020.

All patients showed a complete absence of *SMN1*. Among them, five patients had SMA type 1, and another four patients had SMA type 2. Each patient with SMA type 1 had two copies of *SMN2*, and each patient with SMA type 2 had three copies of *SMN2*.

Together with patients who were born during 2011–2015 [16], details of patients with infantile SMA who were born in Shikoku during the period 2011–2020 are summarized in Table 1.

### 3.3. Incidence of Infantile SMA between 2016 and 2020

As shown in Table 2, the SMA incidence during the period 2016–2020 was 7.08 (95% CI 2.45–11.71) per 100,000 live births. According to our previous report [16], the SMA incidence was 2.70 per 100,000 live births (95% CI 0.05–5.35) during the period 2011–2015. When the two datasets were combined, the incidence during 2011–2020 was 4.73 per 100,000 live births (95% CI 2.16–7.30). It should be noted that in our series, no patients had an intragenic mutation in *SMN1*, suggesting the extreme rarity of such cases.

## 4. Discussion

### 4.1. Increased SMA Incidence in Shikoku

The results of recent surveys of SMA are summarized in Table 3 [6,16,18,19,20]. Verhaart et al. reported that the incidence of SMA was approximately 11.9 in 100,000 live births [7]. In Osaka and Hyogo prefectures in Japan, Kimizu et al. reported that the incidence of all SMA types and SMA type 1 was 3.1 and 1.3 in 100,000 live births [6]. Ito et al. estimated that the incidence of SMA was 4.2 in 100,000 live births in a nationwide survey in Japan [21]. The reported incidence in Japan was relatively lower than that reported in other countries.

Incidence of SMA indicates the diagnosis rate of the disease. Diagnosis is made based on the disease criteria and, finally, genetic testing. However, it might be incorrect to interpret this as meaning that the lower incidence is owing to the very strict diagnostic criteria applied in patients with SMA or limited availability of genetic testing in Japan. Instead, the lower incidence might be related to low awareness about SMA among the Japanese population.

In the present study, we showed that the incidence of infantile SMA in Shikoku was 7.08 per 100,000 live births during the period 2016–2020. The incidence seemed much higher than 2.70 per 100,000 live births during 2011–2015, although these two values are considered to be within the range of variability.

We attributed the increased incidence to introduction of the new, more effective drugs nusinersen and onasemnogene abeparvovec in Japan during 2016–2020. The news that effective drugs were available for an incurable disease, SMA, was widely publicized in Japan. The increased incidence may reflect the increase in clinicians’ awareness about SMA in Shikoku, leading to earlier diagnosis of the disease.

In Shikoku, NBS programs were started in October 2021 in some districts of Ehime Prefecture [22]. Although no cases were identified via NBS in this study, the widespread implementation of NBS may increase the detection rate of patients with SMA in Shikoku.

### 4.2. New Therapeutic Drugs and NBS Programs

SMA has been considered an incurable disease. However, treatments for this disease are emerging at an incredible rate. The United States Food and Drug Administration (FDA) approved nusinersen in 2016, onasemnogene abeparvovec in 2019, and risdiplam in 2020. The Japanese Ministry of Health, Labour and Welfare approved nusinersen in 2017, onasemnogene abeparvovec in 2020, and risdiplam in 2021. Clinical trials of these new drugs show improved prognoses and motor function in infants affected by SMA [10,11,12].

More recently, treatment with nusinersen or onasemnogene abeparvovec at the pre-symptomatic stage in the neonatal period has shown maximum effects on the improvement of motor function. Some infants who were expected to have SMA type 1 based on genetic analysis were able to walk owing to pre-symptomatic treatment with nusinersen or onasemnogene abeparvovec [13,15].

The purpose of NBS is to enable affected infants to be identified and treated with these new drugs as early as possible. Detection of SMA via NBS and treatment of the infant before symptom onset (or before massive loss of motor neurons) could prevent severe delay in the development of motor function or life-threatening respiratory failure. Additionally, normal achievement of motor milestones can be expected. As shown in Table 1, six of 13 patients in our study underwent tracheostomy owing to severe respiratory insufficiency. If these patients had been treated with the new drugs at earlier stage, tracheostomy could have been prevented.

Neonatal screening has made it possible to treat pre-symptomatic patients with SMA. However, there is debate as to whether the copy number of *SMN2* should be used to select candidates for such early treatment [23]. The *SMN2* copy number is a strong prognostic predictor, but prediction is not absolute (as mentioned in the Introduction). Additionally, measurement of the copy number is not always accurate [24]. Although algorithms of treatment remain controversial, early treatment is preferable even with four copies of *SMN2* [24].

### 4.3. SMA Surveys and NBS Programs

The results of many SMA surveys combined with NBS programs have been reported recently (Table 4) [9,25,26,27,28,29,30,31]. A total of 288 newborns with SMA were detected out of 3,674,277 (7.84 in 100,000) infants in an NBS survey conducted in nine countries [8]. SMA surveys in some areas with NBS programs may show more precise results than surveys in other areas without an NBS program. However, the number of areas where an NBS program is implemented is limited.

It is also necessary to keep in mind several limitations of NBS itself. Some infants with SMA might die before being screened for SMA. Some families might refuse to have their infant tested because NBS for SMA is a type of genetic testing. Additionally, current NBS for SMA can only detect the absence of *SMN1* exon 7; NBS does not detect any intragenic *SMN1* mutations.

At present, we cannot determine whether NBS directly influences the incidence of SMA in Japan. However, it is our hope that NBS for SMA will become widespread worldwide because these programs will certainly increase the number of patients with SMA who are treated earlier with newly available drugs.

## 5. Conclusions

The incidence of infantile SMA in Shikoku during the period 2016–2020 was 7.08 (95% CI 2.45–11.71) per 100,000 live births, which was much higher than that during 2011–2015 (2.70; 95% CI 0.05–5.35). These results suggest that the introduction of new therapeutic drugs in Japan led to an increase in clinicians’ awareness and earlier diagnosis of SMA in Shikoku.

## Figures and Tables

**Table 1 IJNS-08-00052-t001:** Patients’ clinical information.

Case	Birth Year	Sex	Subtype	Copy Number		Genetic Testing	Onset (Month)	Diagnosis (Month)	Respiratory Support ^e^
				*SMN1*	*SMN2*				
1 ^a^	2012	Male	1	0	2	qPCR	2	4	TPPV
2 ^a^	2013	Male	1	0	2	qPCR	1	4	TPPV
3 ^a^	2014	Male	1	0	3 ^d^	qPCR	9	13	No
4 ^a^	2015	Female	1	0	2	MLPA	6	8	TPPV
5	2016	Male	2	0	3	MLPA	13	23	No
6	2017	Male	1	0	2	MLPA	2	3	TPPV
7 ^b^	2017	Female	1	0	2	MLPA	3	4	NIPPV
8 ^c^	2018	Male	1	0	2	MLPA	2–3	3	TPPV
9 ^c^	2018	Male	1	0	2	MLPA	2–3	3	TPPV
10	2018	Male	2	0	3	MLPA	9	15	No
11	2019	Female	1	0	2	MLPA	3	5	No
12	2020	Male	2	0	3	MLPA	8	11	No
13	2020	Female	2	0	3	MLPA	10	14	No

^a^ Cases 1–4 were reported in our previous study [15]; ^b^ The patient was born in Shikoku, and then moved out of Shikoku; ^c^ Patients 8 and 9 are twins; ^d^ This case was a patient with type 1 who had three copies of *SMN2*, suggesting the inverse correlation rule is not absolute (See the Introduction section); ^e^ In each case, facilities for respiratory support were available at the research point. Abbreviations: qPCR, real-time quantitative PCR; polymerase chain reaction; MLPA, multiplex ligation-dependent probe amplification, TPPV, tracheostomy with invasive positive-pressure ventilation, NIPPV, non-invasive positive-pressure ventilation.

**Table 2 IJNS-08-00052-t002:** Patients with infantile SMA during 2011–2020 on Shikoku.

Year	Type 1	Type 2	All Types	Live Births	Incidence
2011	0	0	0	30,798	
2012	1	0	1	30,301	
2013	1	0	1	29,687	
2014	1	0	1	28,661	
2015	1	0	1	28,503	
2016	0	1	1	27,546	
2017	2	0	2	26,975	
2018	2	1	3	25,786	
2019	1	0	1	23,901	
2020	0	2	2	22,884	
2011–2015	4	0	4	147,950	2.70 ^a^
2016–2020	5	4	9	127,092	7.08 ^a^
2011–2020	9	4	13	275,042	4.73 ^a^

^a^ Incidence per 100,000 live births.

**Table 3 IJNS-08-00052-t003:** Spinal muscular atrophy incidence reported in previous studies.

Country/Region	Period	Live Births	All Types		Type 1		Reference
			Cases	Incidence ^a^	Cases	Incidence ^a^	
Greece	1995–2018	2,437,348	200	8.2			Kekou, 2020 [18]
Estonia	1996–2020	347,993	42	12.1	24	6.9	Sarv, 2021 [19]
Japan	2007–2016	1,197,156	37	3.1	21	1.3	Kimizu, 2021 [6]
Europe	2011–2015	22,325,221	3776	16.9			Verhaart, 2017 [20]
Japan	2011–2015	147,950			4	2.7	Okamoto, 2019 [16]

^a^ Incidence per 100,000 live births.

**Table 4 IJNS-08-00052-t004:** Incidence of spinal muscular atrophy determined via newborn screening.

Country/Region	Period	Live Births	Cases Number		Incidence ^a^		References
			All Subtypes	*SMN2*, 2 Copies	All Subtypes	*SMN2*, 2 Copies	
Taiwan	November 2014–September 2016	120,267	7	3	5.82	2.49	Chien, 2017 [25]
Japan	January 2018–April 2019	4157	0	0	-	-	Shinohara, 2019 [26]
Germany	January 2018–June 2019	213,279	30	-	14.07	-	Czibere, 2020 [27]
Australia		103,903	9	-	8.66	-	Kariyawasam, 2020 [28]
USA (New York State)	2018	225,093	8	3	3.55	1.33	Kay, 2020 [9]
Southern Belgium	March 2018–February 2021	136,339	10	5	7.33	3.67	Boemer, 2021 [29]
USA (North Carolina)		12,065	1	1	8.29	8.29	Kucera, 2021 [30]
United States (USA)	October 2019–October 2020	60,984	6	2	9.84	3.28	Baker, 2022 [31]

^a^ Incidence per 100,000 live births.

## Data Availability

Detailed clinical information of the study is not shown but is available on request from the corresponding author.

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
