# Peer review of "Changes in the Incidence of Infantile Spinal Muscular Atrophy in Shikoku, Japan between 2011 and 2020"

_2409-515X, 2022, doi:10.3390/ijns8040052_

Round 1

Reviewer 1 Report

The manuscript entitled "Changes in the Incidence of Infantile Spinal Muscular Atrophy in Shikoku, Japan between 2011 and 2020" describe the changes in incidence of infantile SMA in the Shikoku region. The study has been well designed and the results are clearly presented, and discussed. Especially the authors discusse the problem that the published/reportes incidence of a disease is often strongly related to the awareness of a disorder among physicians.

I have only a few remarks and suggestions, which could clarify a few points, that might be not 100% clear.

1. The authors should descibe the nature and function of SMN2 in more detail in the introduction, with appropriate citations.

2. page 2 lines55-59: The authors correctly describe that the severity of SMA is related to the SMN2 copy numbers, and that "in general" patients with SMA type 1 have 2 SMN2 copies, and SMA type 2 have three. However, this could lead the reader to the incorrect assumption that there is a 100% predictive value of SMN2 copy numbers for the severity of SMA, and with that a very good predictor. But this is not the case, and can be seen in table 1, case 3 (SMA type 1 with 3 SMN2 copies). The author should add a sentence to make this clear in the introduction, and also add something in the discussion to point out the problems that might turn up, when introducing NBS for SMA. Screening relies on SMN1 copy numbers, which will not only detect the infantile subtypes, but also subtype 4.

3. page 2 lines60-75: The authors describe the world wide incidence and the propect of NBS. The authors should include in the introduction and also the discussion the fact, that approx. 96% of SMA cases are homozygous for a 5q-associated deletion, and at the moment only those ar detectable by real-time PCR in NBS. Therefore approx. 4% of SMA case will not be detected by NBS.

4. In view of point 3, it would be extremly interesting, to see the genetic results of the 13 patients from table 1. This should be included in the table, or if not available, mentioned in the text.

Author Response

Response to Reviewer 1

[General comments]

The manuscript entitled "Changes in the Incidence of Infantile Spinal Muscular Atrophy in Shikoku, Japan between 2011 and 2020" describe the changes in incidence of infantile SMA in the Shikoku region. The study has been well designed and the results are clearly presented, and discussed. Especially the authors discuss the problem that the published/reported incidence of a disease is often strongly related to the awareness of the disorder among physicians.

I have only a few remarks and suggestions, which could clarify a few points, that might be not 100% clear.

[Response to the general comments]

Thank you very much for your positive feedback.

We very much appreciate the reviewer’s understanding of the core of our paper, that the incidence of the disease could be changed with greater awareness about SMA among physicians.

Thank you very much again for your helpful comments regarding our manuscript. We have revised the manuscript according to your suggestions. Our point-by-point responses are given below.

[Comments 1 and 2; Description about SMN2 copy numbers]

  1. The authors should describe the nature and function of SMN2 in more detail in the introduction, with appropriate citations.
  2. page 2 lines55-59: The authors correctly describe that the severity of SMA is related to the SMN2 copy numbers, and that "in general" patients with SMA type 1 have 2 SMN2 copies, and SMA type 2 have three. However, this could lead the reader to the incorrect assumption that there is a 100% predictive value of SMN2 copy numbers for the severity of SMA, and with that a very good predictor. But this is not the case, and can be seen in table 1, case 3 (SMA type 1 with 3 SMN2 copies). The author should add a sentence to make this clear in the introduction, and also add something in the discussion to point out the problems that might turn up, when introducing NBS for SMA. Screening relies on SMN1 copy numbers, which will not only detect the infantile subtypes, but also subtype 4.

[Answer to Comments 1 and 2]

We have provided a more detailed description regarding copy numbers of SMN2 in the introduction, and added some relevant citations.

(1) Introduction section

We have revised the sentences in the second paragraph, as below:

Page 2, lines 56–63

  • The number of copies of survival motor neuron 2 (SMN2) is related to severity of the disease. More copies of SMN2 is related to an improved prognosis 2,4,5. In general, nearly all patients with SMA type 1 have one or two copies of SMN2. Most patients with SMA type 2 have three copies of SMN2, and most with SMA type 3 have three or four copies of SMN22,5,6. Patients with SMA type 4 usually have four copies or more 2. However, this inverse correlation is not absolute; some patients with two copies of SMN2 have milder phenotypes whereas some with three copies of the gene have been described as having SMA type 1 5.

(2) Table 1

We have added a description of Case 3 in Table 1.

  • d This case was a patient with type 1 who had three copies of SMN2, suggesting the inverse correlation rule is not absolute (See the Introduction section).

(3) Discussion section

We added a new paragraph at the end of Section 4.2. New therapeutic drugs and NBS programs.

Page 6, line 255–260

  • Neonatal screening has made it possible to treat pre-symptomatic patients with SMA. However, there is debate as to whether the copy number of SMN2 should be used to select candidates for such early treatment 23. The SMN2 copy number is a strong prognostic predictor, but prediction is not absolute (as mentioned in the Introduction). Additionally, measurement of the copy number is not always accurate24. Although algorithms of treatment remain controversial, early treatment is preferable even with four copies of SMN224.

Newly added references

  1. Glascock, J.; Sampson, J.; Haidet-Phillips, A.; Connolly, A.; Darras, B.; Day, J.; Finkel, R.; Howell, R. R.; Klinger, K.; Kuntz, N.; et al. Treatment Algorithm for Infants Diagnosed with Spinal Muscular Atrophy through Newborn Screening. J. Neuromus-cul. Dis. 2018, 5, 145–158. https://doi.org/10.3233/JND-180304.
  2. Glascock, J.; Sampson, J.; Connolly, A. M.; Darras, B. T.; Day, J. W.; Finkel, R.; Howell, R. R.; Klinger, K. W.; Kuntz, N.; Prior, T.; et al. Revised Recommendations for the Treatment of Infants Diagnosed with Spinal Muscular Atrophy Via Newborn Screening Who Have 4 Copies of SMN2. J. Neuromuscul. Dis. 2020, 7, 97–100. https://doi.org/10.3233/JND-190468.

[Comments 3 and 4; Description about patients retaining SMN1 gene]

  1. page 2 lines60-75: The authors describe the world wide incidence and the prospect of NBS. The authors should include in the introduction and also the discussion the fact, that approx. 96% of SMA cases are homozygous for a 5q-associated deletion, and at the moment only those ar detectable by real-time PCR in NBS. Therefore approx. 4% of SMA case will not be detected by NBS.
  2. In view of point 3, it would be extremly interesting, to see the genetic results of the 13 patients from table 1. This should be included in the table, or if not available, mentioned in the text.

[Answer to Comments 3 and 4]

You pointed out the lack of a description regarding intragenic mutation in SMN1.

The occurrence of intragenic mutation in SMN1 is very rare (found in only 4%–5% of patients with SMA). Thus, we only used the SMN1 deletion for the purpose of epidemiological analysis. As you suggested in your comments, we should describe this in the Introduction section as well as in the Results and Discussion sections. Thus, following your suggestion, we have revised the manuscript, as below.

(1) Introduction

We have added a sentence (underlined) with citations in the second paragraph. We also have added a sentence (underlined) in the third paragraph.

Page 2, lines 55–56

  • The remaining patients carry an intragenic mutation in the retaining SMN1 3,4.

Newly added references

  1. Wijaya, Y. O. S.; Ar, Rohmah, M.; Niba, E. T. E.; Morisada, N.; Noguchi, Y.; Hidaka, Y.; Ozasa, S.; Inoue, T.; Shimazu, T.; Takahashi, Y.; et al. Phenotypes of SMA patients retaining SMN1 with intragenic mutation. Brain. Dev. 2021, 43, 745-758. https://doi.org/10.1016/j.braindev.2021.03.006.

Page 2, lines 69–71

  • It should be noted that the current NBS program does not include detection of an intragenic mutation in SMN1 because the occurrence of intragenic mutation in SMN1 is very rare.

(2) Table1

The genetic results of the 13 patients were included in table 1.

(3) Results

We have added a sentence at the end of Section 3.3. Incidence of infantile SMA between 2016 and 2020.

Page 4, lines 186–187

  • It should be noted that in our series, no patients had an intragenic mutation in SMN1, suggesting the extreme rarity of such cases.

The layout of some tables might seem to be broken due to the system or Microsoft Word's problems.

Reviewer 2 Report

- Lines 54-55: Sentence "After the discovery of SMN1, SMA came to be diagnosed 54 using genetic testing; muscle biopsy is no longer a common diagnostic procedure" (lines 54-55) is out of place in this paragraph. 

-Line 60: Diagnosis is easier now with genetic testing. Maybe reframe to discuss that difficulty in diagnostics relates to lack of awareness? 

-Line 68: SMA is still incurable (change 'was' to 'is')

-Lines 79-81: I am not sure if this is the place to address it, but it does bring up a topic that perhaps should be addressed. This paper suggests that low incidence of SMA previously was attributed to lack of awareness for the disorder, however, if that is the case those babies obviously still exist. Is there a way to meaningfully look at unexplained infant death numbers from 2011-2015 then from 2016-2020 to see if a shift in those statistics are observed? 

-Line 84: I like the idea of defining this as diagnosis rate. Could this be incorporated earlier in the discussion?

-Table 1: Was respiratory support part of the diagnosis of the child? Or just note of current treatment? Can you probe more into what prompted the diagnosis for these babies and perhaps address the delays still observed in diagnosis based on the table? It may help your argument that increased awareness is contributing to increased diagnostics. It may not be necessary for the paper, but it would be interesting to know. 

Table 2: The 'a' notation only on 2011-2015 numbers; I assume all incidence were per 100,000 births? Also, last line should be 2020-2021, not 2021-2020. 

Author Response

Response to Reviewer 2

Thank you for your helpful comments regarding our manuscript. We have revised the manuscript according to your suggestions. Our point-by-point responses are given below.

[Comment 1; Description of muscle biopsy]

- Lines 54-55: Sentence "After the discovery of SMN1, SMA came to be diagnosed 54 using genetic testing; muscle biopsy is no longer a common diagnostic procedure" (lines 54-55) is out of place in this paragraph.

[Answer to Comment 1]

As you have suggested, we deleted the following sentence. (Page 2, line 54)

  • After the discovery of SMN1, SMA came to be diagnosed using genetic testing; muscle biopsy is no longer a common diagnostic procedure. (Page 2, line 54)

We also deleted the part related to the sentence mentioned above.

  • Additionally, the evidence regarding diagnosis has changed as the molecular genetics of SMA have advanced (Page 2, line 61)

[Comment 2; Genetic testing and difficulty in SMA diagnosis]

-Line 60: Diagnosis is easier now with genetic testing. Maybe reframe to discuss that difficulty in diagnostics relates to lack of awareness? 

[Answer to Comment 2]

Thank you very much for your helpful suggestion.

(1) Introduction

We have omitted the phrase “that is not easy to diagnose” in the original version.

Page 2, line 64

  • SMA is a rare disease, and the incidence of SMA varies among countries 7.

(2) Discussion

We have added the underlined sentence in the new second paragraph of Section 4.1. Increased SMA incidence in Shikoku. The new paragraph has been inserted between the first and third paragraph in the original version.

Page 5, lines 209–213

  • Incidence of SMA indicates the diagnosis rate of the disease. Diagnosis is made based on the disease criteria and, finally, genetic testing. However, it could be wrong to interpret that the lower incidence is owing to the very strict diagnostic criteria applied in patients with SMA or limited availability of genetic testing in Japan. Instead, the lower incidence might be related to low awareness about SMA among the Japanese population.

[Comment 3; SMA is still an incurable disease]

-Line 68: SMA is still incurable (change 'was' to 'is')

[Answer to Comment 3]

As you suggested, we have changed 'was' to 'is'. (page 2, line 72)

[Comment 4; Undiagnosed SMA patients]

-Lines 79-81: I am not sure if this is the place to address it, but it does bring up a topic that perhaps should be addressed. This paper suggests that low incidence of SMA previously was attributed to lack of awareness for the disorder, however, if that is the case those babies obviously still exist. Is there a way to meaningfully look at unexplained infant death numbers from 2011-2015 then from 2016-2020 to see if a shift in those statistics are observed? 

[Answer to Comment 4]

The number of patients with SMA in our study is the number who were diagnosed during the study period. We do not know the outcome of patients who were not diagnosed during the study period. Some of those patients might have died without a correct diagnosis during the study period (including unexplained infant death), Some patients might be alive or have died with delayed diagnosis after the study period and others might be alive or have died without correct diagnosis after the study period.

[Comment 5; Definition of incidence]

-Line 84: I like the idea of defining this as diagnosis rate. Could this be incorporated earlier in the discussion?

[Answer to Comment 5]

Thank you very much for your agreement with the definition of incidence in our study.

To incorporate the definition of incidence earlier in the discussion, we have added a sentence (underlined) in the new second paragraph of Section 4.1. Increased SMA incidence in Shikoku.

Page 5, lines 209–213

  • Incidence of SMA indicates the diagnosis rate of the disease. Diagnosis is made based on the disease criteria and, finally, genetic testing. However, it could be misleading to believe that the lower incidence is owing to the very strict diagnostic criteria applied in patients with SMA or limited availability of genetic testing in Japan. Instead, the lower incidence might be related to low awareness about SMA among the Japanese population.

[Comment 6; Respiratory support of patients]

-Table 1: Was respiratory support part of the diagnosis of the child? Or just note of current treatment? Can you probe more into what prompted the diagnosis for these babies and perhaps address the delays still observed in diagnosis based on the table? It may help your argument that increased awareness is contributing to increased diagnostics. It may not be necessary for the paper, but it would be interesting to know. 

[Answer to Comment 6]

Thank you very much for your comment.

Respiratory support in Table 1 reflects the patient condition during the study period, but this item (treatment) is not directly associated with the diagnostic process.

We have added a description in Table 1.

e In each case, facilities for respiratory support were available at the research point.

The reason we added these data into the table with diagnosis is we would like to emphasize the importance of awareness about SMA leading to early diagnosis and treatment. Thus, we included new text in the third paragraph of Section 4.2. New therapeutic drugs and NBS programs,

Page 5, lines 243–246

  • As shown in Table 1, six of 13 patients in our study underwent tracheostomy owing to severe respiratory insufficiency. If these patients had been treated with the new drugs at earlier stage, tracheostomy could have been prevented.

[Comment 7; Errors in Table 2]

Table 2: The 'a' notation only on 2011-2015 numbers; I assume all incidence were per 100,000 births? Also, last line should be 2020-2021, not 2021-2020. 

[Answer to Comment 7]

Thank you very much for pointing out the errors in Table 2.

We have corrected all errors in the table.

The layout of some tables might seem to be broken due to the system or Microsoft Word's problems.